# Unexpected Genetic Diversity of Two Novel Swine MRVs in Italy

**DOI:** 10.3390/v12050574

**Published:** 2020-05-22

**Authors:** Lara Cavicchio, Luca Tassoni, Gianpiero Zamperin, Mery Campalto, Marilena Carrino, Stefania Leopardi, Paola De Benedictis, Maria Serena Beato

**Affiliations:** 1Diagnostic Virology Laboratory, Department of Animal Health, Istituto Zooprofilattico Sperimentale delle Venezie (IZSVe), Viale dell’Università 10, Legnaro, 35020 Padua, Italy; lcavicchio@izsvenezie.it (L.C.); tassoni.luca@gmail.com (L.T.); mcampalto@izsvenezie.it (M.C.); mcarrino@izsvenezie.it (M.C.); 2OIE Collaborating Centre for Diseases at the Animal/Human Interface, Istituto Zooprofilattico Sperimentale delle Venezie (IZSVe), Viale dell’Università 10, Legnaro, 35020 Padua, Italy; gzamperin@izsvenezie.it (G.Z.); sleopardi@izsvenezie.it (S.L.); pdebenedictis@izsvenezie.it (P.D.B.)

**Keywords:** MRV2, MRV3, swine MRVs, reassortment, novel MRV2 group

## Abstract

Mammalian Orthoreoviruses (MRV) are segmented dsRNA viruses in the family *Reoviridae*. MRVs infect mammals and cause asymptomatic respiratory, gastro-enteric and, rarely, encephalic infections. MRVs are divided into at least three serotypes: MRV1, MRV2 and MRV3. In Europe, swine MRV (swMRV) was first isolated in Austria in 1998 and subsequently reported more than fifteen years later in Italy. In the present study, we characterized two novel reassortant swMRVs identified in one same Italian farm over two years. The two viruses shared the same genetic backbone but showed evidence of reassortment in the S1, S4, M2 segments and were therefore classified into two serotypes: MRV3 in 2016 and MRV2 in 2018. A genetic relation to pig, bat and human MRVs and other unknown sources was identified. A considerable genetic diversity was observed in the Italian MRV3 and MRV2 compared to other available swMRVs. The S1 protein presented unique amino acid signatures in both swMRVs, with unexpected frequencies for MRV2. The remaining genes formed distinct and novel genetic groups that revealed a geographically related evolution of swMRVs in Italy. This is the first report of the complete molecular characterization of novel reassortant swMRVs in Italy and Europe, which suggests a greater genetic diversity of swMRVs never identified before.

## 1. Introduction

Mammalian Orthoreoviruses (MRVs) are double-stranded RNA viruses belonging to the *Reoviridae* family, *Spinareovirinae* subfamily, *Orthoreovirus* genus. Viruses belonging to the *Orthoreovirus* genus are divided into ten species based on their host: *Avian orthoreovirus, Baboon orthoreovirus, Broome orthorevorirus, Mahlaptisi orthoreovirus, Mammalian orthoreovirus, Nelson bay orthoreovirus, Neoavian orthoreovirus, Piscine orthoreovirus, Reptalian orthoreovirus, Testudine orthoreovirus* [1]. Without an envelope and having an icosahedral symmetry, their dimensions range between 65–80 nm and are clearly visible under the electron microscope [2]. The MRV genome, of about 23,000 base pairs (bp), consists of ten gene segments divided into three Large (L) of about 3,800 bp, three Medium of about 2,200 bp (M), four Small (S) of about 1,100 bp [3]. They have a characteristic protein profile with three lambda (λ), three mu (µ) and four sigma (σ) primary translation products derived from L, M and S genes respectively, as well as additional small gene products that are encoded by polycistronic segments [2].

*Mammalian orthoreovirus* includes all the nonfusogenic orthoreoviruses, with three major serotypes (MRV-1, MRV-2 and MRV-3) representing numerous isolates, and a fourth serotype with only one isolate, Ndelle reovirus (MRV-4Nd) [4]. Based on the phylogenetic classification of the S1 segment, prototype strains have been identified: Type 1 Lang-T1L, Type 2 Jones-T2J, Type 3 Dearing-T3D and Type 3 Abney-T3A [5]. Amino acid sequence identities of the sigma-class major outer capsid proteins and core proteins of various MRV serotypes range from 90% to 97% (ICTV) [6]. These properties are linked to the S1 gene, encoding a capsid protein (σ1) responsible for the adhesion of the virus to cellular receptors and therefore responsible for tissue tropism [2].

Reassortant MRV strains are frequently detected because of their segmented genome. MRVs have a wide geographical distribution and are theoretically capable of infecting all types of mammals; in particular, they have been detected in pigs, bats and humans, being associated with both asymptomatic and symptomatic infections [2,7,8,9,10,11,12,13,14,15,16,17,18]. However, in recent years, MRVs have often been described as being the only pathogen in human hosts causing severe enteritis, acute respiratory infections and encephalitis [3,7,19,20,21,22]. Recently, reports on human MRVs have increased worldwide, describing human MRVs as reassortant strains involving MRVs detected in the animal reservoirs [19,20,23]. In Europe, a MRV2 was detected in a person presenting gastroenteritis who had returned to Slovenia after travelling from Southeast Asia (Thailand/Myanmar) [19]. A study conducted in Japan on samples collected between 1981 and 2018 showed that MRV2 strains circulating in Japan and in other East Asian countries for at least two decades in humans, causing gastro-enteric diseases [23] presented a high genetic similarity for the S2 gene with bat and swine MRVs (swMRVs) [24].

Furthermore, in 2019 a MRV3 strain was detected in a child with diarrhea in Brazil and, based on the S1 phylogeny, the strain resulted closely related to Asian swMRVs, isolated between 2010 and 2016 and associated with diarrhea in pigs [20].

Regarding MRV in swine, sporadic cases have been reported worldwide since the late ‘90s [8,23]. In 2006 and 2007, MRV3 was detected in pigs in the presence of respiratory and gastro-enteric signs in China [18]. Sequence information are available only for swMRVs detected in China between 2011 and 2013. MRV3 was reported in 2012 and in 2013 in pigs presenting diarrhea in Korea [16]. In 2014, MRV3 was detected in diarrheic pigs in the USA in association with Porcine Epidemic Diarrhea (PED) [17]. One year later, MRV3 was reported for the first time in Europe, more precisely in Italy, where it was associated with PED [13]. The first MRV3 detection in Italy resulted from an accidental identification while attempting virus isolation in cell cultures during routine diagnostic investigations in the diarrheic faeces of PED infected pigs [13].

The role of MRV in diarrheal manifestation in pigs remains uncertain and seems to be in relation to the age of infected animals; however, evidence suggests that MRV may also contribute to the severity of gastrointestinal manifestations [8,9,13,16,18].

Experimental infections in neonatal pigs provided contradictory data regarding the pathogenic role of MRV3 in pigs [16,17]. An experimental study conducted in 2-day old piglets with an American MRV3 showed severe clinical manifestations since day one post infection (p.i.) [17]. By contrast, Qin et al. [16] showed that a Chinese MRV3 possessed a mild pathogenic capacity in piglets of the same age, with absence of diarrhea and vomiting. In addition, a vaccine efficacy study was conducted in sows to evaluate clinical protection in offspring by experimental challenge with the American MRV3 [17,25]. Surprisingly, 3-day old piglets born from unvaccinated sows and subsequently infected did not present any clinical signs. By contrast, experimental infection using 3-day old gnotobiotics piglets resulted only in mild disease [25].

The present report describes the first identification and molecular characterization of two reassortant MRVs belonging to two different serotypes (MRV3 and MRV2) from one single swine farm located in North Eastern Italy. These viruses presented an exceptional molecular gene composition compared to previous swMRVs described in pigs. The phylogenetic analysis indicated that the two novel swMRVs presented a similar genetic composition except for the S1, S4 and M2 gene segments, which showed distinct characteristics and were likely acquired from reassortant events with MRVs from other sources, including bats, humans or other hosts yet to be defined. The genetic diversity was supported by several unique amino acid (aa) signatures on the S1 protein of both viruses. Our study provides novel information and insights into emerging swMRVs, which present reassortments and a remarkable genetic heterogeneity in the swine population in Italy. This suggests that further investigations are needed to better understand the circulation pattern of swMRVs with an undiscovered diversity and a potential zoonotic impact.

## 2. Materials and Methods

### 2.1. MRVs Detection in One Single Swine Farm in North East Italy

In February 2016, an acute episode of gastroenteritis in fattening pigs was reported by the field veterinarian in a closed-cycle farrow to finish farm located in the province of Treviso, Veneto region, in North Eastern Italy. Animals presented acute gastroenteritis with diarrhea and were observed for 5–6 days. No mortality events were reported. Six faeces samples were collected from the diseased animals and submitted to the diagnostic department of the Istituto Zooprofilattico Sperimentale delle Venezie (IZSVe), where they were processed for virus isolation in cell cultures yielding an MRV isolate (case number: 16DIAPD52154/2016).

The IZSVe researchers visited the same swine farm two years later with the aim of re-assessing the circulation of MRV (10 October 2018). Faeces samples (*n* = 8) were collected from healthy pigs (case number: 18DIAPD90178/2018). In particular, two samples were collected from gilts, two from sows, two from porker and two from fattening finisher pigs.

Each faecal sample was diluted 1:5 (*w*/*v*) in Phosphate Buffered Saline (PBS) supplemented with antibiotics (10,000 IU/mL of penicillin G, 10 mg/mL of streptomycin, 5000 IU/mL nystatin, 0.25 mg/mL gentamicin sulphate). Faeces homogenates were diluted in a final 20% glycerol solution (*v*/*v*) (Sigma Aldrich, St. Louis, MO, USA) as a preserving agent, vortexed and centrifuged at +4 °C for 5 min at 14,000× *g*. The supernatant was then aliquoted in 2 mL tubes and stored at −80 °C until use for detection and characterization of MRVs.

Two hundred and eighty microliters (µL) of each faeces homogenate supernatant and 1120 µL of Lysis Buffer (AVL) were used for viral RNA extraction using the “QIAamp Viral RNA mini kit” (QIAGEN, Hilden, Germany) according to the manufacturer’s instructions. The eluted RNA was used for detection of MRV by a one-step RT-PCR targeting a L1 conserved region using primer pair L1-rv5F and L1-rv6R [26] and “One Step RT PCR kit” (QIAGEN, Hilden, Germany). In detail, the L1-rv5F and L1-rv6R primer pair target a conserved fragment of 416 bp of the MRVs RNA-dependent RNA polymerase (RdRp) gene [26].

The one-step RT-PCR was carried out in 25 µL reaction volume consisting of 12.5 µL of water, 5 µL of Buffer, 0.4 µM of each primer, 0.4 µM of dNTPs, 20 U Rnase inhibitor, 1 µL enzyme mix, 2.5 µL of the extracted RNA. After a dsRNA denaturation step, at 95 °C for 5 min; the PCR program used had the following thermal profile: retro-transcription at 50 °C for 30 min; a Taq activation step at 95 °C for 15 min followed by 45 cycles with denaturation at 94 °C for 30 s, annealing at 52 °C for 30 s, extension at 72 °C for 40 s and a final elongation at 72 °C for 5 min. The PCR products were analysed by 7% acrylamide gel followed by a silver staining.

For MRV characterization, one-step RT-PCRs targeting the S1 gene of serotype 2 (S1.2) and 3 (S1.3) were applied. One-step RT-PCR targeting the S1.2 was performed using the SuperScript III One-Step RT-PCR System with Platinum Taq DNA polymerase kit (Invitrogen, Carlsbad, CA, USA) and ad hoc designed primers: FOR WGBR (5′-TGCTAGAGTCACAGCCCT-3′) and REV WGBR (5′-AATTCCTTGTTCTGTAGCAT-3′). The One-step RT-PCR was carried out in 25 µL reaction volume consisting of 6 µL of water, 12.5 µL of Buffer, 0.2 µM of each primer, 0.5 µL enzyme mix, 5 µL of the extracted RNA. After a dsRNA denaturation step at 95 °C for 5 min, the PCR program had the following thermal profile: retro-transcription at 50 °C for 20 min; Taq activation at 95 °C for 2 min, followed by 40 cycles with denaturation at 95 °C for 15 s, annealing at 45 °C for 30 s, extension at 72 °C for 30 s and a final elongation at 72 °C for 5 min. The One-step RT-PCR targeting the S1.3 was performed using the “One Step RT PCR kit” (QIAGEN, Hilden Germany) and the S1R3 primers described by Leary [26]. The One-step RT-PCR was carried out in 25 µL reaction volume consisting of 12.5 µL of water, 5 µL of Buffer, 0.4 µM of each primer, 0.4 µM of dNTPs, 20U Rnase inhibitor, 1 µL enzyme mix, 2.5 µL of the extracted RNA. After a dsRNA denaturation step at 95 °C for 5 min, the PCR program had the following thermal profile: retro-transcription at 50 °C for 30 min; Taq activation at 95 °C for 15 min, followed by 45 cycles with denaturation at 94 °C for 30 s, annealing at 54 °C for 30 s, extension at 72 °C for 40 s and a final elongation at 72 °C for 5 min. The PCR products were analysed by 7% acrylamide gel followed by a silver staining.

### 2.2. Virus Isolation in Cell Cultures

Virus isolation was attempted for all samples collected in 2016 and 2018, using African green monkey kidney (Vero) cells (ATCC number CCL-81) grown in Minimal Essential Medium (MEM) (Gibco, Whaltham, MA, USA) with 0.25 µg/mL amphotericin and 0.05 µg/mL gentamicin.

Virus isolation was carried out as described by Chen et al. [27]. Three blind passages were performed.

Negative staining Electron Microscopy (EM) [28], RT-PCR targeting the L1 of the MRVs [26] and typing RT-PCRs targeting the S1.2 and S1.3 genes [26] were carried out on cell cultures presenting cytopatic effect (CPE).

### 2.3. Reconstruction of MRV Genomes

Next generation sequencing (NGS) was carried out on two viral isolates, one from 2016 and one from 2018. Purified total RNA was subjected to retrotranscription with SuperScript III Reverse Transcriptase (Invitrogen, Carlsbad, CA, USA) and NEBNext mRNA Second Strand Synthesis Module (Euroclone, Stockholm, Sweden) or with Maxima H Minus Double-stranded cDNA synthesis kit (ThermoFisher, Carlsbad, CA, USA). Double-stranded cDNA was purified with Agencourt AMPure XP (Beckman Coulter, Brea, CA, USA) and quantified with Qubit dsDNA HS assay kit (ThermoFisher, Carlsbad, CA, USA). The cDNA library was prepared using Illumina Nextera XT DNA Sample Preparation kit (Illumina, San Diego, CA, USA), and fragments were selected with Agencourt AMPure XP (Beckman Coulter, Brea, CA, USA). Library was checked for quality and size with Agilent 2100 Bioanalyzer (Agilent High Sensitivity DNA kit, Agilent Technologies, Carlsbad, CA, USA), and sequenced with Miseq v2 or v3 Reagent Kit (250 or 300PE) using Illumina MiSeq platform.

Illumina reads quality was assessed using FastQC v0.11.2 (A quality control tool for high throughput sequence data: Babraham Bioinformatics web site). Raw data were filtered by removing: (i) reads with more than 10% of undetermined (“N”) bases; (ii) reads with more than 100 bases with Q score below 7; (iii) duplicated paired end reads. Remaining reads were clipped from Illumina adaptors Nextera XT with scythe v0.991 (Available on line: https://github.com/vsbuffalo/scythe, accessed on 25/02/2019) and were trimmed with sickle v1.33 (Avalable on line: https://github.com/najoshi/sickle, accessed on 25/02/2020). Reads shorter than 80 bases or unpaired after previous filters were discarded.

The taxonomic assignment of high-quality reads was carried out using BLAST v2.6.0+ or v2.7.1+ alignment [29] against the integrated NT database (version 8 February 2017 or 12 February 2018) and diamond v0.8.36 or v0.9.17 alignment against the integrated NR database (version 8 February 2017 or 12 February 2018) [30]. Alignment hits with e-values higher than 1 × 10^−3^ were discarded. The taxonomic level of each read was determined by the lowest common ancestor (LCA) -based algorithm implemented in MEGAN Ultimate Edition v6.7.0 or v6.10.8. For the MRV genome assembly, high-quality reads taxonomically classified as belonging to the *Reoviridae* family were selected [31].

High-quality *Reoviridae* reads were further divided into ten groups, each one corresponding to one specific MRV segment, by (i) aligning them with BLAST to the integrated NT database, (ii) manually assigning them to each group based on their best hit and iii) checking the uniqueness of all assignments. Each group was then separately de novo assembled using IDBA-UD version 1.1.1, with default parameters and the multi-k-mer approach (minimum value, 24; maximum value, 124; increment, 5 or 10) [32]. For each assembly, the longest contig with a length comparable to the expected MRV segment genome size was selected. All reads used in a specific assembly were subsequently aligned against the corresponding selected contig using BWA version 0.7.12, with standard parameters [33]. The alignment was manually revised with Tablet v1.14 to (i) check all used reads fell into the assembly, (ii) verify that all nucleotides (nt) were the consensus ones, (iii) verify the absence of misaligned reads, and (iv) avoid the risk of misassembly [34].

Finally, since genome termini of some swMRV segments was poorly sequenced in NGS due to the expected declining profile of coverage in these regions, the genome termini were Sanger sequenced with a primer walking approach. The amplification of each terminal gene segment was carried out using external primers, designed on untranslated regions (UTRs) of each segment described by Wang et al. [35], and internal primers designed on obtained NGS sequences (Appendix A). Briefly, one-step RT-PCR was carried out using “One Step RT PCR kit” (QIAGEN, Hilden, Germany). After a dsRNA denaturation step at 95 °C for 5 min, the PCR program used had the following thermal profile: retro-transcription at 50 °C for 30 min, Taq activation step at 95 °C for 15 min followed by 35 cycles with denaturation step at 94 °C for 30 s, annealing for 30 s with a different temperature according to the melting temperature of primer pair used, extension at 72 °C for 40 s and a final elongation at 72 °C for 5 min. The sequencing was conducted with the Ab3130xl instrument (ThermoFisher, Carlsbad, CA, USA). Raw data were elaborated with Seqscape v2.6 and the obtained sequences compared with those available in Genbank using BLAST (Available at: https://blast.ncbi.nlm.nih.gov/Blast.cgi, accessed on 13 February 2020).

### 2.4. MRV Phylogenetic Analysis

The assumed evolutionary relationships between the two identified swMRVs were reconstructed by phylogenetic analysis using nt sequences of the ten gene segments.

MRV nt sequences included in the phylogenetic trees were retrieved using the two Italian MRV sequences as BLAST queries. The dataset formed by all the available BLAST results was further expanded including the results obtained by keywords research in Genbank.

The obtained datasets were aligned using MAFFT online software version 7 (Available on line: https://mafft.cbrc.jp/alignment/software/ accessed on 15 January 2020) and the alignment was codon-aligned and edited manually using MEGA 6.0 [36].

The maximum-likelihood phylogenetic trees were inferred by IqTree v1.6.1 [37] using the ModelFinder implemented in the software [38] and performing a non-parametric bootstrap with 100 replicates to obtain the branch support. The trees obtained were edited using Figtree v1.4.3 (Available on line: http://tree.bio.ed.ac.uk/software/figtree/ accessed on 15 January 2020) and the figures were prepared using Adobe Illustrator, removing bootstrap values below 60%.

The two genomes obtained in this study were compared with known MRVs from Genbank included in the dataset to evaluate genetic similarity. The p-distances of the nt and aa sequences were calculated using MEGA 6.0. The presence of unique aa mutations of the S1 sequences generated in the present study was investigated. The MRV genome sequences were deposited in GenBank under accession numbers: MT151659, MT151661, MT151663, MT151666, MT151668, MT151670, MT151672, MT151674, MT151676, MT151678 (MRV3) and MT151660, MT151662, MT151664, MT151665, MT151667, MT151669, MT151671, MT151673, MT151675, MT151677 (MRV2).

### 2.5. MRV2 and MRV3 S1 Homology Modelling

The homology modelling of the S1 protein of the identified swMRVs was carried out to identify the position of aa mutations on a three-dimensional structure of the S1 protein. The deduced linear aa sequences of investigated swMRVs were used. The homology modelling of the three dimensional structures of the proteins was performed by Swissmodel [39,40] homology-modelling server. The obtained structures were visualized using Protein Imager [41] and the unique mutations identified in the molecular analysis were highlighted.

## 3. Results

### 3.1. Identification of Two Novel MRVs in One Single Swine Farm in North Eastern Italy

RT-PCR for MRV on faeces samples yielded positive results for five out of six (5/6) samples of cases 16DIAPD52154/2016 (No. 1–4 and 6) and for five out of eight (5/8) samples of case 18DIAPD90178/2018 (No. 3, 5, 6, 7, 8). Typing RT-PCR on the S1 gene characterized all five MRV positive faeces samples of case 16DIAPD52154/2016 as MRV3 and two out of five (2/5) MRV positive faeces samples of case number 18DIAPD90178/2018 (No. 5, 8) as MRV2. The S1 molecular characterization yielded inconclusive results for three samples of case number 18DIAPD90178/2018 (No. 3, 6, 7).

All (*n* = 5) MRV3 faecal samples of case 16DIAPD52154/2016 and five out of eight (5/8) MRV2 faecal samples of case number 18DIAPD90178/2018 (No. 1, 3, 5, 6, 8) were successfully isolated in cell cultures (Appendix A). Interestingly, one faeces sample negative by RT-PCR resulted positive by virus isolation (No. 1) and, by contrast, one faeces sample positive by RT-PCR proved to be negative by isolation (No. 7). Virus isolation attempts yielded positive results for MRV subsequently confirmed also by EM and RT-PCR for both cases.

All five MRV positive samples of case 16DIAPD52154/2016 presented CPE during the second, third or fourth passages (Appendix A). Four out of five MRV positive faeces samples of case 18DIAPD90178/2018 presented CPE during the second passage, although all five cell cultures tested positive by RT-PCR (Appendix A). Direct electron microscopy examination confirmed the presence of *Reoviridae* particles (Appendix A).

Two viruses, one per each case, were selected to infect Vero cells and produce virus stocks, showing CPE at 24 h and 48 h p.i., respectively (Figure 1A,B). The viruses were designated as: MRV3/swine/Italy/52154-4/2016 (MRV3/swine/Ita/2016) and MRV2/swine/Italy/90178-3/2018 (MRV2/swine/Ita/2018) for further analysis.

### 3.2. NGS Sequencing of Swine MRV2 and MRV3

We employed a metagenomic approach to retrieve the complete genome of the two selected swMRV strains, MRV3/swine/Ita/2016 and MRV2/swine/Ita/2018 from the same swine farm in North Eastern Italy to avoid any target-amplification bias. Through NGS sequencing we produced 4,768,352 paired-end 251 bp reads for MRV3 and 6,680,153 paired-end 301 bp reads for MRV2. After quality filtering we kept 78% (MRV3) and 90% (MRV2) of the sequenced data; among these high-quality data, for both samples the MRV read fraction is about 0.05%, which corresponds to a 33X and 67X fold coverage for the MRV3 and MRV2 genome, respectively. All segments of both strains showed the same level of sequence coverage, with a good evenness along the entire genome. MiSeq raw data were submitted to the NCBI Sequence Read Archive (SRA) under the accession numbers SRR11243336-SRR11243338.

### 3.3. Molecular Characterization of Swine MRV2 and MRV3

Sequence analysis yielded 10 contigs for each isolate showing similarity to the previously characterized swMRVs. The ends of all gene segments of MRV2/swine/Ita/2018 were successfully Sanger sequenced designing specific primer pairs.

Swine MRV3/swine/Ita/2016 represents the second detection of swine MRV3 in Italy and Europe, elapsing one year, and the third worldwide. Swine MRV2 represents the second identification both in Europe and globally after remaining undetected for 21 years.

Interestingly, the two novel swMRVs were identified in the same swine farm over a two-year period: MRV3 in February 2016 and MRV2 in September 2018.

Comparisons of MRV3/swine/Ita/2016 and MRV2/swine/Ita/2018 across the whole genome showed a common genetic backbone, with high nt similarity in the L1, L2, L3, M1, M3, S2, S3 gene segments, ranging from 98.57% to 99.69%, increasing at the aa level (99% to 99.90%) (Table 1, Appendix A). On the other hand, the S1, S4 and M2 genes of the novel Italian swMRVs identified displayed a high degree of nt diversity with the MRV3/swine/Ita/2015 and with all the available MRVs sequences (Table 1, Figure 2 and Figure 3), conferring uniqueness to the newly identified swMRVs.

The topology of the nt phylogenetic trees revealed that segments of the common genetic backbone (Appendix A), except for M1 (Appendix A), formed distinct genetic clusters of swMRVs. Moreover, four of the segments composing the genetic backbone (L1, L3, M1 and M3) clustered with the Italian MRV identified in swine in 2015 (MRV3/swine/Ita/2015, Accession numbers: KX343200.1, KX343202.1, KX343203.1, KX343205.1) (Appendix A). Furthermore, three of these four gene segments, L1, L3 and M3, formed a monophyletic clade of Italian swMRVs only (Appendix A). Besides the evidence provided by the phylogenetic tree topology, the low genetic distances with other swMRVs also supported the strong relation between the three Italian swMRVs identified so far (Table 1).

The phylogenetic analyses based on the S1 segment classified the swMRV detected in 2016 as serotype 3 and the one detected in 2018 as MRV2 (Figure 2), which supports the assumption of a within-farm reassortment between 2016 and 2018, which changed the genetic composition of the 2018 swMRV strain.

In detail, the phylogenetic analysis of MRV3/swine/Ita/2016-S1 showed a high nt similarity (>98%) with four MRV strains of bat and human origins isolated between 2010 and 2015 in Slovenia, Italy and Switzerland (Table 1, Figure 2A).

By contrast, MRV2/swine/Ita/2018-S1 gene presented low nt similarity (<90%) with all the MRV2-S1 sequences available so far, which proved its great diversity to the point of being considered as a novel reovirus (Table 1). Interestingly, MRV2/swine/Ita/2018-S1 clustered with MRV detected from a swine in Taiwan in 2015 (LC482244.1), from a common vole in Hungary in 2006 (KX384852.1) and from a bat detected in Slovenia in 2008 (MG457114.1), with nt similarity of 88.55%, 85.19% and 84.89%, respectively (Table 1) and aa similarity between 87.58% and 88.91% (Table 1) and forming a unique distinct genetic cluster (Figure 2B). The identification of such a distinct MRV2-S1 cluster is further underpinned by the nt differences ranging between 60% and 71% from all the other MRV2 strains and by nt similarity between MRV1-S1 and MRV2-S1 strains, which ranges between 51% and 59% (data not shown). Although MRVs belonging to this newly identified genetic cluster present an intragroup nt similarity between 84.1% and 93%, they share fourteen unique aa mutations that can be considered as distinctive signatures for this group (Table 2).

Interestingly, both strains seemed peculiar for S4 and M2 segments, having less than 94% nt similarity with all the MRVs described so far (Figure 3, Table 1). The low nt similarities observed for the novel Italian swMRVs were reflected at the aa level only for the M2 segment but not for the S4 (Table 1).

In detail, the S4 segments of MRV3/swine/Ita/2016 and MRV2/swine/Ita/2018 presented a low nt similarity (93.52%) and clustered with two Chinese viruses collected in 2011 and 2012 (Figure 3A), although sharing with them a low nt similarity: 92.81–94.44% and 94.37–94.80% with the Italian 2016 and 2018 swMRV strains, respectively (Table 1). Of note, the majority of the swMRV-S4 gene segments identified starting from year 2014 formed a distinct genetic cluster, supported by high bootstrap values, including viruses from China, Italy, Taiwan and the USA (Figure 3A). Interestingly, a subgrouping within this cluster could be identified with viruses from China and Italy forming one subgroup, those from Taiwan a second one and the American MRVs a third additional one (Figure 3A). Such genetic clustering appeared to have evolved from swine MRV3 isolated in the USA and may suggest a host species restriction of swMRVs (Figure 3A) for this gene segment.

Similarly, the M2 gene segment of the MRV3/swine/Ita/2016 and MRV2/swine/Ita/2018 have shown to be distantly related (nt similarity of 84.10%) (Figure 3B). Furthermore, the M2 segments of the two novel Italian swMRVs did not share any similarity above 93% with any other M2 segment available (Table 1).

The topology of the phylogenetic tree showed that the MRV3/swine/Ita/2016-M2 formed a small and distinct genetic group with MRV1 strains of human origin and prototype MRV1 strains, although sharing a low similarity (92.20–92.57%) (Table 1, Figure 3B).

The MRV2/swine/Ita/2018-M2 gene segment presented the highest nt similarity with a MRV isolate identified from a common vole in Hungary in 2006 (KX384850.1) (90.65%) (Table 1, Figure 3B), indicating its genetic novelty.

Interestingly, the three available M2 sequences from Italian swMRVs (2015, 2016 and 2018) seemed to be distantly related (Figure 3B) and no high nt similarity M2 sequences were observed for any of them (Table 1).

### 3.4. Analysis of the S1 Protein of Swine MRV3 and MRV2

To further analyse the molecular characteristics of the newly identified Italian swMRVs, we compared the deduced aa sequences with those available in Genbank. Four unique aa mutations were identified and distributed on the MRV3/swine/Ita/2016 S1 protein, namely: A87V, F218L, S293P, I335T, the last two being present on the exposed part of the protein and here described for the first time (Figure 4A).

The aa substitutions A87V and S293P encoded for amino acids with the same characteristics, respectively nonpolar and polar uncharged. In addition, the F218L mutation introduced a residue with a different steric hindrance, from an aromatic to a non-polar one, while the I335T mutation introduced a polar, uncharged aa instead of a nonpolar one. Regarding the position of the unique aa mutations, both the S293P and I335T appeared in the exposed part of the protein, as suggested by the S1 homology modelling (Figure 4A). The MRV3/swine/Ita/2016 S1 protein shared some important mutations with other MRVs, which confer protease resistance 249I [42] and an enhanced neurotropism 340D and 419n [43]. In addition, the receptor sialic binding domain (NLAIRLP) of the novel MRV3 strain was conserved [44].

Interestingly, comparing the deduced aa sequences of the swine MRV2 available in the public databases, the MRV2/swine/Ita/2018 presented fourteen unique aa mutations on the S1 protein (Table 3, Figure 4B), the majority of which affected the tail and one the protein head (367S) (Figure 4B).

In addition to these unique aa mutations, MRV2/swine/Ita/2018 presented other mutations shared by all the MRV2 strains belonging to the same genetic cluster characterized by fourteen aa mutations (Table 2), exclusively present in the following strains: MRV2/swine/Italy/90178-3/2018, MRV_T2/microtus_arvalis/Hungary/2006, MRV2/myotis_myotis/Slovenia/SI-MRV05/2008 and MRV2/swine/Taiwan/sR1590/2015. Of these fourteen aa changes, five were characterized by the substitution with a Leucine (L), three with Glycine (G) and two with Serine (S) (Table 2). Eight out of fourteen (8/14) aa mutations implicated the substitution of a polar aa with a nonpolar one. In addition, mutations in position 25 and 172 proved to be peculiar to the MRV2-S1 new genetic cluster identified (Table 2), as in such positions all the other MRV2 presented constantly the same aa. Most of the shared aa mutations by the new MRV2-S1 genetic cluster are located in the first N-terminal half, while no shared mutation was located in the protein head.

## 4. Discussion

The combination of various laboratory diagnostic approaches has made possible to detect two distinct swMRVs circulating in one same swine farm in North Eastern Italy over a two-year period, between 2016 and 2018. A metagenomic approach allowed us to retrieve the sequences of two novel reassortant swMRVs. The MRV2/swine/Ita/2018 sequence was further successfully elongated by an implemented Sanger sequencing approach.

The first MRV identification (MRV3) in the studied swine farm occurred in 2016; the second (MRV2) two years later during a re-assessment sampling to confirm MRV circulation. The novel MRV3 and MRV2 strains identified in the present study showed a remarkably different genetic constellation from all available swMRVs, resulting from reassortant events involving the S1, S4 and M2 genes with the remaining genes that showed a common genetic backbone for both swMRV strains. Therefore, the S1, S4 and M2 genes conferred the genetic uniqueness to both swMRVs identified.

The identification of two different swMRVs (serotype 2 and 3) presenting a similar genetic backbone constituted by seven out of ten (7/10) gene segments suggested a prolonged circulation of MRV, as well as single or multiple introductions of MRVs donating three genes: S1, S4, M2.

The swine MRV2 characterized in the present study represented the first report in Italy and the second in Europe after 21 years from the first description. Of note, so far only five swine MRV2 strains had been reported worldwide: one from Japan in 1994, whose sequences are not available [8], three from Taiwan dating back to 2015 and one from Austria from year 1998, for which only deposited sequences are available. Such paucity in the detection of MRV2 in swine can be attributed to a lack of investigation and surveillance in swine and to the possible failure of current diagnostic techniques for a correct detection and characterization of MRV circulating in swine. As a matter of fact, the recent reports on swine and human MRVs have all been accidental; MRV full characterization was indeed possible through a metagenomic approach adopted to diagnose the unexplained observed clinical manifestations [3,13,21,23]. Although the NGS approach remains a powerful investigation and virus discovery tool, laboratory protocols for MRV detection and characterization are still poorly developed and of limited sensitivity. A robust screening Real Time RT-PCR targeting the L1 gene is not currently available and published protocols for MRV characterization based on the S1 gene need to be adjusted, as demonstrated in the present study. However, large scale studies able to increase the sequence data on MRVs will foster the development of diagnostic tools and improve surveillance measures. So far, such studies have only involved some bat species [11,13,15,45,46,47] but never swine and/or humans, although recent increasing evidence shows MRV circulation also in these latter species [9,12,13,16,19,20,23,25,48,49,50].

MRV2/swine/Ita/2018 appears to be correlated to the MRV3/swine/Ita/2016 for the majority of genes, thus suggesting the persistence of MRV within the farm through intraspecific transmission.

Interestingly, the genes composing the genetic backbone of both novel Italian swMRVs did not present any similarity with MRVs of different host and geographic origins, as for other animal MRVs reported so far. This is therefore the first evidence that a distinct cluster of swMRVs may exist and is possibly evolving in this animal host in Italy. Only four genes, L1, L3, M1 and M3, seemed to have evolved from the first MRV3 detected in swine in Northern Italy in 2015, as evidenced by the tree topology, and therefore considered the putative ancestor for the novel swMRVs reported herein [13]. Nonetheless, this evidence supported the hypothesis of the existence of a great genetic diversity among swMRVs in Italy and geographic driven evolution traits for the novel reassortant Italian swMRVs. To the authors’ knowledge, this is the first report of animal MRVs presenting a specific host and geographic restrictions in contrast to other reports, where no defined host and/or geographic origin barriers have been identified among MRVs [13,15,35,51].

A high similarity between the MRV3/swine/Ita/2016-S1 and one MRV of bat and human origin recently detected in Europe was observed at nt and aa levels [15,52]. Notwithstanding the high nt similarity with other MRV-S1 genes, the MRV3/swine/Ita/2016-S1 presented two unique aa mutations in the exposed part of the protein, described herein for the first time, whose impact on virus attachment, receptor binding avidity and virulence in swine remains unclear and needs further investigation. Mutation I335T affected the head portion of the protein involved in the binding of host cell receptors; further investigations are needed to better clarify the biological effects of this unique mutation and to investigate whether the impact of the two conferring protease resistance and enhanced neurotropism is maintained.

The high similarity of the MRV3/swine/Ita/2016-S1 gene with bat and human strains, as well as its circulation in asymptomatic pigs, poses questions regarding the possible bat/pig interactions and the zoonotic potential of animal MRVs.

In order to better understand the existence of possible interactions between bats and pigs, ecological studies should be encouraged not only in Italy but in Europe as well, particularly in densely populated pig areas.

Differently, the S1 gene of MRV2/swine/Ita/2018 widely differs from the Italian swMRVs detected so far and presents a low similarity only with the MRV2 strains identified in a bat and in a common vole in Europe; this makes it difficult to demonstrate the origin of such gene, although the remaining ones appear to be correlated to MRV3/swine/Ita/2016. However, such evidence may suggest that an intermediate host representing either bats or micro mammals might have been involved in the reassortment events, which could have been all favoured by the environmental resistance of this virus. The fourteen aa mutations observed in the MRV2-S1.2 cluster and shared among swine, bat and vole strains, are a further evidence of the correlation between MRV2/swine/Ita/2018 and the MRVs from a bat and a common vole, supporting the potential role of small mammals as a reservoir. This hypothesis is further corroborated by the similarity, although low, with a virus from common vole for the MRV2/swine/Ita/2018-M2 gene. In this context, bats displayed a wide variety of MRVs from all serotypes [11,15,35,45,47,53], suggesting they may represent a gene source for reassortment likely through environmental contamination. On the other hand, only one study has been performed in voles [10]; therefore, MRV diversity in these species remains mostly unknown. However, voles can be easily found within farms, thus allowing cross-species transmission of pathogens.

The genetic diversity observed for MRV2/swine/Ita/2018-S1 was further supported by the several unique aa mutations identified and here described for the first time. Interestingly, the MRV2/swine/Ita/2018-S1 formed a new and distinct cluster with other four MRV2 of swine, bat and common vole origin, as evidenced by the topology of the phylogenetic tree, which presented a degree of nt genetic diversity comparable to that existing between the MRV1 and MRV2 S1 genes. This observation raises the possibility that such MRV2-S1 cluster may represent a novel group of MRVs. A deeper investigation of the MRV genetic diversity circulating in the Italian swine population may clarify whether the novel MRV2-S1 cluster might represent a new MRV group.

Such diversity poses several questions, mainly whether it either represents the first identification of a vast genetic group circulating in swine in Italy or a single isolated detection. However, the paucity of MRV2 sequences from the animal reservoir allows to formulate only hypotheses on the origin of MRV2/swine/Italy/90178-3/2018.

The unique mutations present in the σ1 protein, of both serotypes, which are not synonymous, especially if present in the head of the protein, may play a fundamental role in virus replication and adaptation. Several studies intended to analyse the function of MRV aa mutations were conducted using laboratory adapted strains [54,55,56]. The absence of experimental data on aa mutations impact using field MRVs compromises any comparison with our data, as well as with the formulation of any hypothesis on the biological significance of the aa mutations identified.

As for the S4 genes of the novel Italian swMRVs, the tree topology indicates that a possible common origin of this segment might exist, notwithstanding their nt genetic diversity. In truth, the topology of the distinct S4 clustering of swMRVs detected worldwide since 2011 suggests that the identified swMRVs might have evolved from the American swMRVs. Furthermore, this specific clustering is characterized by swMRVs and highlights the species-driven evolution of such gene, in contrast with what has been observed so far [9,16,17,18]. The role of the S4 gene in encoding an external capsid protein could explain the selective pressure specific for swine MRVs. The divergence between nt and aa S4 sequences is easily explained by the presence of synonymous mutations, which highlights the fundamental role of this protein. Moreover, the S4 genetic cluster composed solely by swMRVs indicates that such protein might have played a key-role in the host driven evolution, an issue that certainly deserves to be further investigated.

As with the S4 gene, also the origin of the M2 gene of both MRV3/swine/Ita/2016 and MRV2/swine/Ita/2018 cannot be established with confidence. The low similarity of the MRV3/swine/Ita/2016-M2 with old prototype strains was unexpectedly puzzling and highlighted the uniqueness of such novel reassortant swine MRVs. Similarly, to what was observed for the S1, MRV2/swine/Ita/2018-M2 presented the highest nt similarity, although at low levels, with an MRV detected in a common vole and a bat Therefore, it appears that MRVs of different origin acted as gene donors for the novel Italian swMRVs.

Proteins encoded by reassorted genes (S1, S4, M2) are all structural proteins present in the virion surface and linked to the entry, infectivity and virulence. Whether reassortant events for these genes only represent an advantage for the virus is a matter, which has not been assessed, and can neither be ruled out. The insufficiency of MRVs sequences from swine and/or other animal hosts, including humans, draws a line to the identification of robust hypotheses on the origin of such novel swMRV2 strain. It further emerges that MRVs should deserve more attention considering their segmented genome nature. Taken together, our data suggest that the novel Italian swMRVs originated from reassortant events with uncharacterized viruses, which very likely donated the S4 and M2 to both MRV3/swine/Ita/2016 and the S1, S4 and M2 to MRV2/swine/Ita/2018. It is reasonable to assume that multiple reassortant events must have occurred, to the point of generating the novel swine MRV3 and MRV2 reported herein (Figure 5).

The origin of the S1, S4 and M2 (MRV2/swine/Italy/90178-3/2018) and of the S4 and M2 (MRV3/swine/Italy/52154-4/2016) cannot be determined with high confidence relying on the data available so far. Whether these reassortments represent a unique episode or are the result of multiple introductions of diverse MRVs in different moments within the swine farm object of this investigation remains an uncertain matter. However, based on tree topology and nt similarities, some hypotheses on reassortment events and origin might be put forward (Figure 5). The MRV3/swine/Ita/2016 might be the result of a reassortant event involving the American swMRVs, the MRV3/swine/Ita/2015 and the bat origin MRV3 (MRV3/Eptesicus_serotinus/Slovenia/SI-MRV02/2010), although the nt homologies are low with the swine Italian 2015 and swine American MRVs. The S1 gene of the MRV3/swine/Ita/2016 might have derived from the bat (MRV3/Eptesicus_serotinus/Slovenia/SI-MRV02/2010) or from MRV3/swine/Ita/2015 strains. Instead, MRV2/swine/Ita/2018 might have resulted from a reassortment event involving the MRV3/swine/Ita/2016 and common vole MRV2 donating the S1 and M2 genes, although the nt similarity values are extremely low with the MRV2 strains (Figure 5).

The basis of virus reassortment is given by the coinfection of cells with two viruses, to the point that the frequency of coinfection steers the process. However, coinfection alone does not necessarily lead to reassortment. A series of stochastic events participate in this process, such as the prevalence of circulating viral lineages, the likelihood of dual exposure, the spatial dynamics of the viruses involved and replication strategies [57]. In case of *Reoviridae*, which have a high compartmentalized life cycle, reassortment is likely to occur via the merging of heterologous viral inclusions within the coinfected cells. Whether such reassortments impact on the evolution of MRVs is unknown, as uncertain is the assumption that they may confer a selective advantage. In addition, reassortment events likely represent and/or are accompanied by increased virulence for swine; constant monitoring of MRV circulation and characterization may elucidate this aspect.

The biology and epidemiology of MRVs in animal and human hosts in Italy is largely unexplored. The present study provides insights into the exceptional MRVs genetic diversity that may exist. Additional studies based on a more appropriate sampling approach may better define the characteristics of the MRVs population circulating in swine in Italy, potentially elucidating the genetic relatedness and origin of Italian swMRVs. The yet undefined pathological role of MRVs in swine and other animal species has probably created low interest in investigating the biology of animal MRVs. The potential undetected circulation of MRVs in swine in North Eastern Italy, one of the most densely populated pig areas in the country, as well as the ability to reassort with MRVs from different hosts are matters that should not be underestimated. The recent reports on pathogenic human MRVs sharing homologies with animal MRVs have called for a potential zoonotic transmission, which deserves attention and a one-health approach.

## 5. Conclusions

The absence of data on the MRVs distribution and genetic characteristics in the animal reservoir in Europe jeopardises any conclusion regarding the epidemiology of MRV infections in pigs. Recent studies have shed light on the genetic diversity of swMRVs circulating worldwide and we believe this work has contributed to such evidence; however, robust hypotheses cannot be formulated on their origin and distribution. There are several open questions about MRV infections in swine and in the animal reservoir, and the prevalence, distribution and virulence of MRV serotypes in Italy and Europe still remain unknown matters. The present study has tried to highlight the origin of swMRVs, assuming they might have resulted from reassortant events involving several animal hosts, including humans; special attention should be devoted to the possible and potential animal interactions of swine in order to understand the presence of intermediate hosts and explain the potential mechanism of zoonotic transmission.

## Figures and Tables

**Figure 1 viruses-12-00574-f001:**
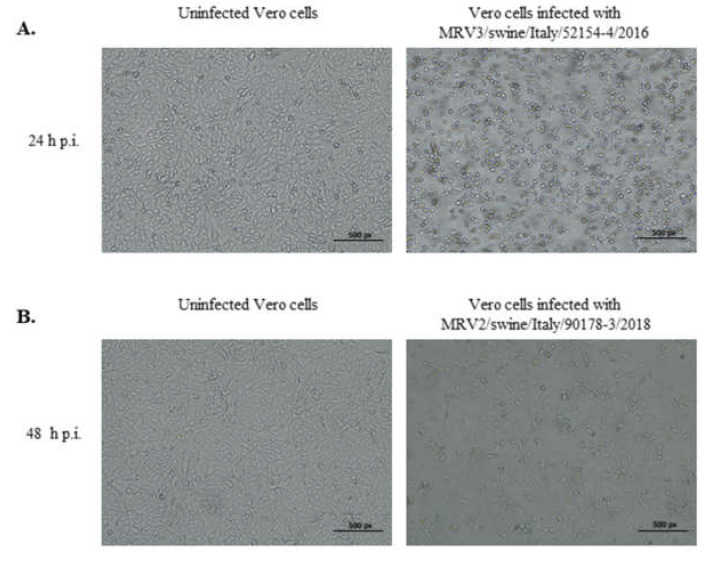
Uninfected Vero cells and CPE observed in Vero cells infected with faeces samples and virus strains. (**A**): MRV3/swine/Italy/52154-4/2016 induced CPE at 24 h p.i. and (**B**): MRV2/swine/Italy/90178-3/2018 induced CPE at 48 h p.i.

**Figure 2 viruses-12-00574-f002:**
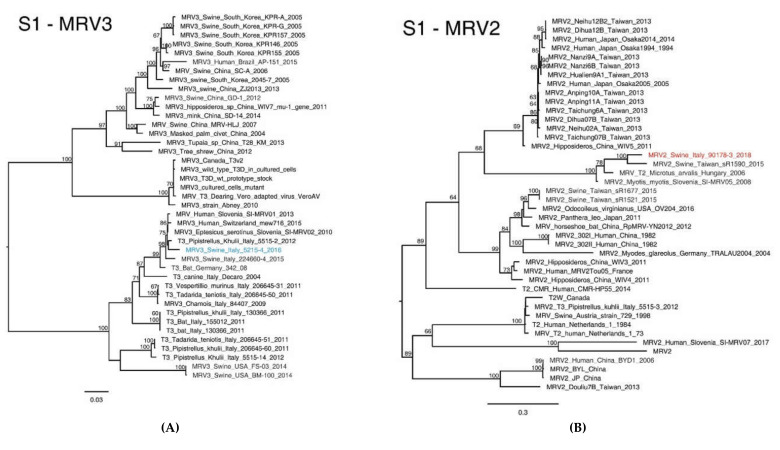
Phylogenetic analysis based on the S1 nucleotide sequence of the MRV3/swine/Italy/52154-4/2016 (**A**) coloured in blue and MRV2/swine/Italy/90178-3/2018 (**B**) strains, coloured in red. The Maximum likelihood phylogenetic tree was obtained using IqTree v1.6.1. The subgroups of the Italian viruses described in this paper are highlighted in different colours. Numbers at the nodes indicate the bootstrap support values. Bootstrap values lower than 60% were omitted.

**Figure 3 viruses-12-00574-f003:**
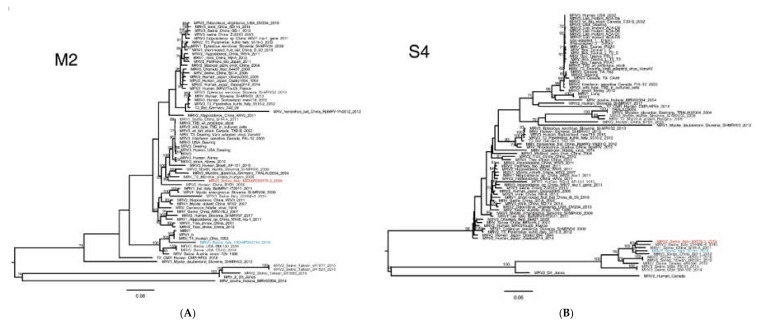
Phylogenetic analysis based on the S4 (**A**) and M2 (**B**) nucleotide sequence of the MRV2/swine/Italy/90178-3/2018 (coloured in red) and MRV3/swine/Italy/52154-4/2016 (coloured in blue) strains. The Maximum likelihood phylogenetic tree was obtained using IqTree v1.6.1. The subgroups of the Italian viruses described in this paper are highlighted in different colours. Numbers at the nodes indicate the bootstrap support values. Bootstrap values lower than 60% were omitted.

**Figure 4 viruses-12-00574-f004:**
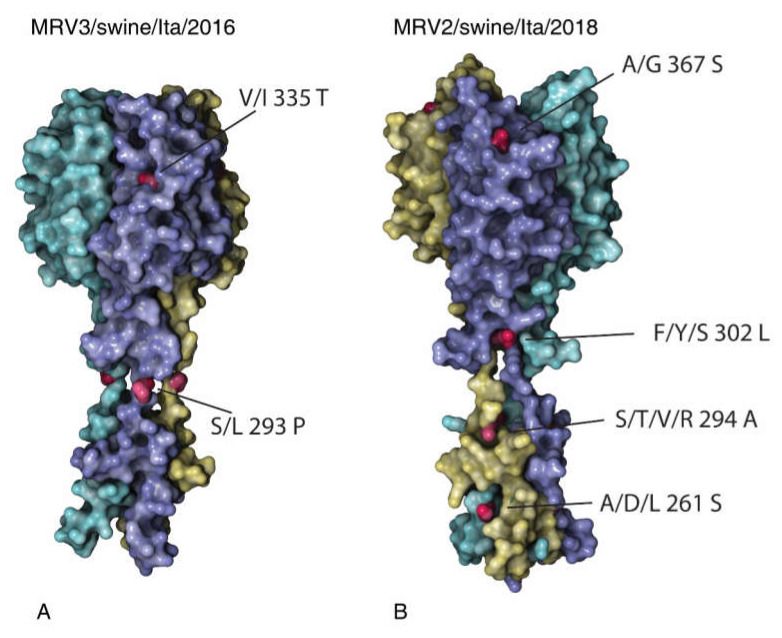
Models of the C-terminal (head domain) of the S1 protein predicted by homology modelling. MRV3/swine/Italy/52154-4/2016 (**A**) and MRV2/swine/Italy/90178-3/2018 (**B**) are shown. Each monomer composing the domain is displayed with a different colour: unique amino acid mutations of each virus are coloured in red.

**Figure 5 viruses-12-00574-f005:**
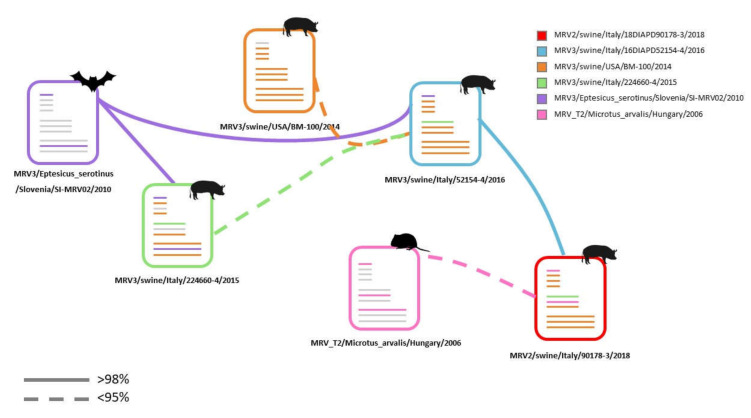
Possible genetic flow scheme based on the nucleotide (nt) similarities and phylogenetic tree topology of swMRVs under study. Dashed line: nt similarity < 95%. Each virus is presented with a specific colour. The grey lines indicate the genes whose origin has not been assessed.

**Table 1 viruses-12-00574-t001:** Nucleotide and amino acid identities for each segment of the MRV3/swine/Ita/2016 and MRV2/swine/Ita/2018 strains, selecting the first three most similar MRV strains.

	MRV3/Swine/Italy/52154-4/2016				MRV2/Swine/Italy/90178-3/2018			
		% nt		% aa		% nt		% aa
**L1**	MRV2/swine/Italy/90178-3/2018	99.57	MRV/RUS/Moscow/2017	100	MRV3/swine/Italy/52154-4/2016	99.57	MRV3/swine/Italy/52154-4/2016	99.91
	MRV3/swine/Italy/224660-4/2015	94.90	MRV3/swine/Italy/90178-3/2018	99.91	MRV3/swine/Italy/224660-4/2015	94.93	MRV3/swine/Italy/224660-4/2015	98.89
	MRV_T3_Dearing_Vero_adapted_virus/VeroAV	92.15	MRV3/swine/Italy/224660-4/2015	98.95	MRV_T3_Dearing_Vero_adapted_virus/VeroAV	92.01	MRV3/swine/USA/FS-03/2014	98.66
**L2**	MRV2_swine/Italy/90178-3/2018	99.40	MRV2_swine/Italy/D1690178-3/2018	99.55	MRV3/swine/Italy/52154-4/2016	99.40	MRV3/swine/Italy/52154-4/2016	99.55
	MRV_T3/murine/France/1961	90.42	MRV_T3/murine/France/1961	97.39	MRV_T3/murine/France/1961	90.21	MRV_T3/murine/France/1961	97.51
	MRV_T1/human/Ohio/1953	89.61	MRV_T1/human/Ohio/1953	96.94	MRV_T1/human/Ohio/1953	89.60	MRV_T1/human/Ohio/1953	97.11
**L3**	MRV2_swine/Italy/90178-3/2018	99.69	MRV2/swine/Italy/90178-3/2018	99.90	MRV3/swine/Italy/52154-4/2016	99.69	MRV3/swine/Italy/52154-4/2016	99.90
	MRV3/swine/Italy/224660-4/2015	94.62	MRV3/swine/USA/FS-03/2014	99.72	MRV3/swine/Italy/224660-4/2015	94.31	MRV1/swine/China/SHR-A/2011	98.81
	MRV1_Lang	93.23	MRV1/swine/China/SHR-A/2011	99.71	MRV1_Lang	92.71	MRV3_chamois/Italy/84407/2009	98.66
**M1**	MRV2_swine/Italy/90178-3/2018	99.24	MRV2/swine/Italy/90178-3/2018	99.01	MRV3/swine/Italy/52154-4/2016	99.24	MRV3/swine/Italy/52154-4/2016	99.00
	MRV2/human/MRV2Tou05/France	91.86	MRV2/human/MRV2Tou05/France	95.31	MRV2/human/MRV2Tou05/France	92.27	MRV2/human/MRV2Tou05/France	96.06
	MRV1/swine/China/SHR-A/2011	89.40	MRV1_T1/human/Ohio/1953	94.74	MRV1/swine/China/SHR-A/2011	89.73	MRV1_T1/human/Ohio/1953	95.38
**M2**	MRV_T1/human/Ohio/1953	92.57	MRV_T1/human/Ohio/1953	98.45	MRV_T2/microtus_arvalis/Hungary/2006	90.65	MRV3/tree_shrew/China/2012	96.57
	MRV1_b	92.52	MRV1_b	98.30	MRV3_human/Abney	89.99	MRV_T2/microtus_arvalis/Hungary/2006	96.57
	MRV1	92.20	MRV1	97.88	MRV1/swine/China/SHR-A/2011	89.90	MRV1/tree_shrew/China/2011	96.42
**M3**	MRV2/swine/Italy/90178-3/2018	98.57	MRV3/swine/Italy/224660-4/2015	96.98	MRV3/swine/Italy/52154-4/2016	98.57	MRV3/swine/Italy/52154-4/2016	99.20
	MRV3/swine/Italy/224660-4/2015	92.11	MRV3/swine/USA/BM-100/2014	96.02	MRV3/swine/Italy/224660-4/2015	91.83	MRV3/swine/Italy/224660-4/2015	97.09
	MRV3/swine/USA/BM-100/2014	90.26	MRV3/swine/USA/FS-03/2014	96.02	MRV3/swine/USA/BM-100/2014	89.75	MRV3/swine/USA/BM-100/2014	96.25
**S1**	MRV3/eptesicus_serotinus/Slovenia/SI-MRV02/2010	98.37	MRV3/eptesicus_serotinus/Slovenia/SI-MRV02/2010	98.45	MRV2/swine/Taiwan/sR1590/2015	88.55	MRV2/swine/Taiwan/sR1590/2015	88.91
	T3/pipistrellus_Khulii/Italy/5515-2/2012	98.23	T3/pipistrellus_Khulii/Italy/5515-2/2012	98.45	MRV_T2/microtus_arvalis/Hungary/2006	85.19	MRV2/myotis_myotis/Slovenia/SI-MRV05/2008	87.80
	MRV/human/Slovenia/SI-MRV01/2013	98.15	MRV/human/Slovenia/SI-MRV01/2013	98.23	MRV2/myotis_myotis/Slovenia/SI-MRV05/2008	84.89	MRV_T2/microtus_arvalis/Hungary/2006	87.58
**S2**	MRV2/swine/Italy/90178-3/2018	98.86	MRV2/swine/Italy/90178-3/2018	99.51	MRV3/swine/Italy/52154-4/2016	98.86	MRV3/swine/Italy/52154-4/2016	99.51
	MRV2/swine/Taiwan/sR1590/2015	93.18	MRV2/odocoileus virginianus/USA/OV204/2016	99.02	MRV_1_Lang_Prototype	93.24	MRV2/odocoileus virginianus/USA/OV204/2016	99.04
	MRV2/swine/Taiwan/sR1677/2015	92.53	MRV1/tree shrew/China/2011	99.02	MRV2/swine/Taiwan/sR1590/2015	92.76	MRV1/tree shrew/China/2011	99.04
**S3**	MRV2/swine/Italy/90178-3/2018	99.52	MRV2/swine/Italy/90178-3/2018	99.41	MRV3/swine/Italy/52154-4/2016	99.52	MRV3/swine/Italy/52154-4/2016	99.41
	T3/bovine/Maryland/clone18/1961	94.00	MRV3/human/Brazil/AP-151/2015	98.55	T3/bovine/Maryland/clone18/1961	94.37	MRV3/swine/USA/FS-03/2014	98.90
	MRV3/swine/USA/FS-03/2014	93.71	MRV3/swine/USA/FS-03/2014	98.25	T1/human/Wash.D.C/clone62/1957	94.10	MRV3/human/Brazil/AP-151/2015	98.66
**S4**	MRV3/swine/China/GD-1/2012	94.80	MRV2/swine/Italy/90178-3/2018	98.09	MRV1/swine/China/SHR-A/2011	94.44	MRV3/swine/Italy/52154-4/2016	98.09
	MRV1/swine/China/SHR-A/2011	94.37	MRV1/swine/China/SHR-A/2011	97.77	MRV3/swine/Italy/224660-4/2015	94.08	MRV1/swine/China/SHR-A/2011	98.08
	MRV3/swine/Italy/224660-4/2015	93.63	MRV3/swine/China/GD-1/2012	97.45	MRV3/swine/Italy/52154-4/2016	93.52	MRV3/swine/Italy/224660-4/2015	97.26

**Table 2 viruses-12-00574-t002:** Amino acid signatures of the MRV2/swine/Ita/2018 S1 protein cluster. ID strains of the S1 genetic cluster are reported in red. The nucleotide and amino acid numbering are according to MRV_T2/microtus_arvalis/Hungary/2006. Viruses in red belong to the same genetic cluster of the MRV2/swine/Ita/2018 strain.

Nucleotide Positions	37 39	73 75	76 78	112 114	331 333	346 348	388 390	406 408	460 462	472 474	514 516	649 651	739 741	799 801
**Amino acid position**	**13**	**25**	**26**	**38**	**111**	**116**	**130**	**136**	**154**	**158**	**172**	**217**	**247**	**267**
MRV2/swine/Italy/18DIAPD90178-3/2018	**F**	**G**	**L**	**L**	**V**	**L**	**G**	**S**	**L**	**Q**	**L**	**A**	**G**	**S**
MRV_T2/microtus_arvalis/Hungary/2006	**F**	**G**	**L**	**L**	**V**	**L**	**G**	**S**	**L**	**Q**	**L**	**A**	**G**	**S**
MRV2/myotis_myotis/Slovenia/SI-MRV05/2008	**F**	**G**	**L**	**L**	**V**	**L**	**G**	**S**	**L**	**Q**	**L**	**A**	**G**	**S**
MRV2/swine/Taiwan/sR1590/2015	**F**	**G**	**L**	**L**	**V**	**L**	**G**	**S**	**L**	**Q**	**L**	**A**	**G**	**S**
MRV2/swine/Taiwan/sR1677/2015	L	E	I	N	A	V	N	N	S	G	T	S	S	N
MRV2/swine/Taiwan/sR1521/2015	L	E	I	N	A	V	N	N	S	G	T	S	S	N
MRV2/human/Japan/Osaka2014/2014	L	E	I	T	T	I	N	T	A	S	T	S	S	N
MRV2/human/Japan/Osaka2005/2005	L	E	I	T	T	I	N	T	A	S	T	S	S	N
MRV2/human/Japan/Osaka1994/1994	L	E	I	T	T	I	N	T	A	S	T	S	S	N
MRV2/hipposideros/China/WIV5/2011	L	E	I	N	T	I	N	T	A	S	T	S	S	N
MRV2/hipposideros/China/WIV3/2011	L	E	I	N	A	V	N	T	G	G	T	S	S	N
MRV/horseshoe_bat/China/RpMRV- YN2012/2012	L	E	I	N	A	V	N	N	N	G	T	S	S	N
MRV2/Nanzi9A/Taiwan/2013	-	-	-	T	T	I	N	T	V	S	T	S	S	N
MRV2/Taiwan/2013 (10 strains)	-	-	-	T	T	I	N	T	A	S	T	S	S	N
MRV2/panthera_leo/Japan/2011	L	E	I	N	T	V	N	N	N	G	T	S	S	N
MRV2/odocoileus_virginianus/USA/OV204/2016	L	E	I	N	T	V	N	N	S	G	T	S	S	N
MRV2/hipposideros/China/WIV4/2011	L	E	I	N	T	V	K	N	G	D	T	S	S	N
MRV2/human/MRV2Tou05/France	L	E	T	N	T	V	N	T	G	G	T	S	S	N
MRV2/myodes_glareolus/Germany/TRALAU2004/2004	L	E	I	N	A	I	R	N	D	D	T	-	S	N
MRV2_302I/human/China/1982	L	E	I	N	A	V	N	N	S	G	T	S	S	N
Most common amino acid in NCBI available sequences	L	E	I	N	T	I	N	T	A	S	T	S	S	N
Other amino acid variants in NCBI available sequences	I	-	V,T	T,A,S	A,S,G	V,R,T	T, E, K, D, R, S	N, A	S, Q, G, N, V, D	G, A, V, D	-	N	A	Q

**Table 3 viruses-12-00574-t003:** Unique amino acid mutations of the MRV2/swine/Ita/2018 S1 protein.

Nucleotide Position *	Amino Acid Position	Unique Amino Acid Mutations of MRV2/Swine/Ita/2018	Typical Amino Acids of MRV2 Strains
1–3	1	L	M
226–228	76	V	T,L,A
241–243	81	I	S,A
355–357	119	S	D,N,V
361–363	121	V	S,T,A,L,I
478–480	160	G	V,D,N,S,A
571–573	191	G	N,T
589–591	197	D	N,S,G,A,R
640–642	214	L	F
655–657	219	V	M,I,L
781–783	261	S	A,D,L
880–882	294	A	S,T,V,R
904–906	302	L	F,Y,S
1099–1101	367	S	G,A

* Numbering according to S1 gene—MRV_T2/microtus_arvalis/Hungary/2006 (Accession number: KX384852.1).

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
