# Peer review of "Unexpected Genetic Diversity of Two Novel Swine MRVs in Italy"

_viruses, 2020, doi:10.3390/v12050574_

Round 1

Reviewer 1 Report

Cavicchio et al. nicely present the identification of two novel sw MRVs in one farm in Northeast Italy.

I have some minor comments.  

Introduction

Page 1, line 33 – there are 10 Orthoreovirus species, not seven. Please, see the current ICTV classification.

Page 2, lines 50-58: a lot of repetitions. I suggest to delete the sentence “…MRVs can cause asymptomatic, respiratory or gastro-enteric infections and, …”

Results Section 3.1

271-280: The description of virus isolation is a bit confusing. Maybe prepare an informative table instead.

For further reading it would be convenient, if you mention at the end of this section the designation of the two selected isolates (MRV3/swine/Ita/2016 and MRV2/swine/Ita/2018). Five MRVs from serotype 2 and 3 were identified, but only two were selected for further characterisation, correct?

Table 1 is confusing. Include only one strain with the highest identity. If possible, merge nucleotide and amino-acid identities in one table side by side as in Feher et al. 2016 or Naglic et al. 2018.

Figure 2: please, add the scale-bar.

Remove Figure 3, as it is already presented in the Supplement. Check the colours. Not only Italian isolates are highlighted in different colours as noted in figure caption. Besides, there are too many colours. It’s confusing. Try to present results with less colours. Maybe highlight only the two novel isolates from this study. Check also other figures in the manuscript (Figure 4 and Figure 5)

Lines 362, 402, 512, 513, 522, 552, 579: The Hungarian MRV isolate described in Feher et al. (2016) is from common vole (not bank vole). Check throughout the manuscript.

Line 490: reference missing

Figure 7: What does grey line designate? For better contrast, change colours of either isolate MRV3/swine/USA/BM-100/2014 or isolate MRV3/swine/Italy/52154-4/2016. Highlight the two isolates from this study. Swap pictures of animals for MRV2/swine/Italy/90178-3/2018 and MRVT2/Microtus_arvalis/Hungary/2006.

Line 569: Add full designation of MRV 2 and MRV3

Line 578: define which bat

Author contribution:

Remove last sentence as it is duplicated.

Reference: Check for duplicates and reference style. 

Author Response

Dear Editor,

We would like to thank the reviewers for their thoughtful comments and efforts towards improving the quality of our manuscript. Following reviewer number 2 suggestions, we are now submitting a revised version of our paper as a short communication and we have moved some information into the supplementary material.

We have carefully addressed each comment and we hope that the revised version is now suitable for publication.

Each comment is in bold, followed by the authors response.

Reviewer 1

(x) English language and style are fine/minor spell check required 

Response:

The revised manuscript has been proofread by a native English speaking colleague.

Cavicchio et al. nicely present the identification of two novel sw MRVs in one farm in Northeast Italy.

I have some minor comments.  

Introduction

Page 1, line 33 – there are 10 Orthoreovirus species, not seven. Please, see the current ICTV classification.

Response:

Correction in line 33 has been made, modifying seven into ten and adding in lines 33-36: Broome orthorevorirus, Neoavian orthoreovirus, Testudine orthoreovirus.

Page 2, lines 50-58: a lot of repetitions. I suggest to delete the sentence “…MRVs can cause asymptomatic, respiratory or gastro-enteric infections and, …”

Response:

We agree with the reviewer and the sentence (lines 55-56) has now been removed.

Results Section 3.1

271-280: The description of virus isolation is a bit confusing. Maybe prepare an informative table instead.

Response:

The Authors have clarified this section (lines 359-365) and moved some details into the supplementary information.

For further reading it would be convenient, if you mention at the end of this section the designation of the two selected isolates (MRV3/swine/Ita/2016 and MRV2/swine/Ita/2018). Five MRVs from serotype 2 and 3 were identified, but only two were selected for further characterisation, correct?

 Response:

The designation of the viruses has now been moved to line 514; in addition, it has been modified to present results more clearly.

Sentence in lines 501-5134 has been modified as well in order to render the information more understandable.

Table 1 is confusing. Include only one strain with the highest identity. If possible, merge nucleotide and amino-acid identities in one table side by side as in Feher et al. 2016 or Naglic et al. 2018.

Response:

The Table has been modified as suggested and presented in lines 642. The comparison of nt and aa similarities are now side by side and only the tree with the most similar viruses are included.

Figure 2: please, add the scale-bar.

 Response:

The Authors have added the scale-bar in the figure, which is now in the  supplementary material section, as suggested by reviewer number 2.

Remove Figure 3, as it is already presented in the Supplement. Check the colours. Not only Italian isolates are highlighted in different colours as noted in figure caption. Besides, there are too many colours. It’s confusing. Try to present results with less colours. Maybe highlight only the two novel isolates from this study. Check also other figures in the manuscript (Figure 4 and Figure 5)

Response:

Figure 3 has been removed. Regarding the corresponding figures in the supplementary material, the two novel viruses have been highlighted using only two colours. The same changes have been made for Figures 2 and 2, presented in the new version in lines 653 and 659.

Lines 362, 402, 512, 513, 522, 552, 579: The Hungarian MRV isolate described in Feher et al. (2016) is from common vole (not bank vole). Check throughout the manuscript.

Response

Common vole has been substituted and checked throughout the manuscript.

Line 490: reference missing

Response:

References number 15 and 52 have been added in line 986.

Figure 7: What does grey line designate? For better contrast, change colours of either isolate MRV3/swine/USA/BM-100/2014 or isolate MRV3/swine/Italy/52154-4/2016. Highlight the two isolates from this study. Swap pictures of animals for MRV2/swine/Italy/90178-3/2018 and MRVT2/Microtus_arvalis/Hungary/2006.

Response:

Figure 7 has been modified and is now Figure 5. In detail: the designation of the grey line has been clarified in lines 1107-1108.

Line 569: Add full designation of MRV 2 and MRV3

Response:

The full designation has been added (lines 1109-1110).

Line 578: define which bat

 Response:

The full designation has been added in line 1119.

Author contribution:

Remove last sentence as it is duplicated.

 Response:

Removed, see line 1181.

Reference: Check for duplicates and reference style. 

Response:

The Authors have added the references using Refworks. However, a careful check has been made and modifications have been made in order to comply with the correct style.

Reviewer 2 Report

The chosen subject is important to widen our knowledge about the circulating strains and the evolutinary mechanisms of orthoreoviruses. I would recommend to present the results in the form of a short communication and move several unnecessary descriptions, basic methods, results (CPE, EM, etc) into the Supplementary material. In figures and tables using several closely related strains, going into such details is irrelevant and reduces transparency.

Author Response

Dear Editor,

We would like to thank the reviewers for their thoughtful comments and efforts towards improving the quality of our manuscript. Following reviewer number 2 suggestions, we are now submitting a revised version of our paper as a short communication and we have moved some information into the supplementary material.

We have carefully addressed each comment and we hope that the revised version is now suitable for publication.

Each comment is in bold, followed by the authors response.

Reviewer 2

 (x) Extensive editing of English language and style required 

Response:

The revised manuscript has been proofread by a native English speaking colleague.

Yes

Can be improved

Must be improved

Not applicable

Does the introduction provide sufficient background and include all relevant references?

( )

(x)

( )

( )

Are the results clearly presented?

( )

(x)

( )

( )

Response:

The introduction and results section have been modified to improve clarity

Comments and Suggestions for Authors

The chosen subject is important to widen our knowledge about the circulating strains and the evolutinary mechanisms of orthoreoviruses. I would recommend to present the results in the form of a short communication and move several unnecessary descriptions, basic methods, results (CPE, EM, etc) into the Supplementary material. In figures and tables using several closely related strains, going into such details is irrelevant and reduces transparency.

Response:

The paper has been submitted in the revised version as a Short Communication. Authors have followed the reviewer’s suggestion and moved some details previously in the material and methods and result sections into the supplementary material section.

Paragraphs 3.1. and 3.3 have been modified to make the results clearer (lines 477-515; 625-629). Figures and tables have been modified as suggested by reviewer 1 as well, and we have decreased the number of strains used for comparison (see revised Table 1). Table 2 has been modified as well, grouping some information in the last two raws. In addition, Figures 1A, 1B, 2 have been moved into the supplementary information.

The introduction has been modified as well, increasing the clarity.